# HIV status alters disease severity and immune cell responses in Beta variant SARS-CoV-2 infection wave

Farina Karim[1,2], Inbal Gazy[2,3], Sandile Cele[1,2], Yenzekile Zungu[1], Robert Krause[1,2], Mallory Bernstein[1], Khadija Khan[1,2], Yashica Ganga[1], Hylton Rodel[1,4], Ntombifuthi Mthabela[1], Matilda Mazibuko[1], Daniel Muema[1,2], Dirhona Ramjit[1], Thumbi Ndung'u[1,4,5,6], Willem Hanekom[1,4], Bernadett Gosnell[7], COMMIT-KZN Team, Richard J Lessells[2,3,8], Emily B Wong[1,9], Tulio de Oliveira[2,3,8,10,11], Mahomed-Yunus S Moosa[7], Gil Lustig[8], Alasdair Leslie[1,2,4]*, Henrik Kløverpris[1,2,4,12]*, Alex Sigal[1,2,6]*

[1]Africa Health Research Institute, Durban, South Africa; [2]School of Laboratory Medicine and Medical Sciences, University of KwaZulu-Natal, Durban, South Africa; [3]KwaZulu-Natal Research Innovation and Sequencing Platform, Durban, South Africa; [4]Division of Infection and Immunity, University College London, London, United Kingdom; [5]HIV Pathogenesis Programme, The Doris Duke Medical Research Institute, University of KwaZulu-Natal, Durban, South Africa; [6]Max Planck Institute for Infection Biology, Berlin, Germany; [7]Department of Infectious Diseases, Nelson R. Mandela School of Clinical Medicine, University of KwaZulu-Natal, Durban, South Africa; [8]Centre for the AIDS Programme of Research in South Africa, Durban, South Africa; [9]Division of Infectious Diseases, Department of Medicine, University of Alabama at Birmingham, Birmingham, United States; [10]Centre for Epidemic Response and Innovation, School of Data Science and Computational Thinking, Stellenbosch University, Stellenbosch, South Africa; [11]Department of Global Health, University of Washington, Seattle, United States; [12]Department of Immunology and Microbiology, University of Copenhagen, Copenhagen, Denmark

*For correspondence:
Al.Leslie@ahri.org (AL);
Henrik.Kloverpris@ahri.org (HK);
alex.sigal@ahri.org (AS)

Group author details:
COMMIT-KZN Team See page 13

Competing interests: The authors declare that no competing interests exist.

**Abstract** There are conflicting reports on the effects of HIV on COVID-19. Here, we analyzed disease severity and immune cell changes during and after SARS-CoV-2 infection in 236 participants from South Africa, of which 39% were people living with HIV (PLWH), during the first and second (Beta dominated) infection waves. The second wave had more PLWH requiring supplemental oxygen relative to HIV-negative participants. Higher disease severity was associated with low CD4 T cell counts and higher neutrophil to lymphocyte ratios (NLR). Yet, CD4 counts recovered and NLR stabilized after SARS-CoV-2 clearance in wave 2 infected PLWH, arguing for an interaction between SARS-CoV-2 and HIV infection leading to low CD4 and high NLR. The first infection wave, where severity in HIV negative and PLWH was similar, still showed some HIV modulation of SARS-CoV-2 immune responses. Therefore, HIV infection can synergize with the SARS-CoV-2 variant to change COVID-19 outcomes.

## Introduction

HIV is a prevalent infection in KwaZulu-Natal, South Africa (**Kharsany et al., 2018**) which also has a high SARS-CoV-2 attack rate (**Tegally et al., 2021a**; **Tegally et al., 2021b**). HIV depletes CD4 T helper cells (**Dalgleish et al., 1984**) which are a critical part of the adaptive immune response and

are also the main target of HIV infection. CD4 T cell death occurs after cellular infection with HIV (*Westendorp et al., 1995*), or in bystander or incompletely infected cells due to activation of cellular defense programs (*Doitsh et al., 2010*; *Doitsh et al., 2014*), and is halted and, to some extent, reversed by antiretroviral therapy (ART), even sub-optimal therapy (*Jackson et al., 2018*).

The loss of CD4 T cells leads to dysregulation of many aspects of the immune response, including germinal center formation and antibody affinity maturation, which requires help from the highly HIV susceptible CD4 T follicular helper cells (*Okoye and Picker, 2013*; *Pallikkuth et al., 2012*; *Perreau et al., 2013*). In association with this, HIV also causes B cell dysregulation and dysfunction (*Moir and Fauci, 2013*). Moreover, T cell trafficking, activation, and exhaustion profiles of both CD4 and CD8 subsets are also modulated by HIV infection (*Day et al., 2006*; *Deeks et al., 2004*; *Mavigner et al., 2012*).

Both antibody and T cell responses are critical for effective control and clearance of SARS-CoV-2. More severe COVID-19 disease correlates with lymphopenia and low T cell concentrations (*Lucas et al., 2020*; *Sekine et al., 2020*; *Chen et al., 2020a*), whilst mild disease correlates with a robust T cell response to SARS-CoV-2 (*Grifoni et al., 2020*; *Sekine et al., 2020*; *Rydyznski Moderbacher et al., 2020*; *Mathew et al., 2020*; *Mateus et al., 2020*; *Liao et al., 2020*; *Chen et al., 2020b*). Neutralizing antibodies and associated expansion of antibody secreting B cells (ASC) are elicited in most SARS-CoV-2 infected individuals (*Woodruff et al., 2020*; *Robbiani et al., 2020*; *Quinlan et al., 2020*), and neutralizing antibody titers strongly correlate with vaccine efficacy (*Khoury et al., 2021*; *Earle et al., 2021*), indicating their key role in the response to SARS-CoV-2 infection. In contrast, high neutrophil numbers are associated with more severe disease and an elevated neutrophil to lymphocyte ratio (NLR) is often considered a risk factor for a more severe COVID-19 outcome (*Liu et al., 2020a*; *Liu et al., 2020b*; *Zhang et al., 2020*).

Results from epidemiological studies of the interaction between HIV and SARS-CoV-2 from other locations are mixed. Several large studies observed that disease severity and/or mortality risk is increased with HIV infection (*Western Cape Department of Health in collaboration with the National Institute for Communicable Diseases, South Africa et al., 2021*; *Geretti et al., 2021*; *Bhaskaran et al., 2021*; *Tesoriero et al., 2021*; *Braunstein et al., 2021*; *Jassat et al., 2021a*) while others found no statistically significant differences in clinical presentation, adverse outcomes, or mortality (*Huang et al., 2021*; *Sigel et al., 2020*; *Shalev et al., 2020*; *Vizcarra et al., 2020*; *Stoeckle et al., 2020*; *Dandachi et al., 2021*; *Härter et al., 2020*; *Karmen-Tuohy et al., 2020*; *the Northwell COVID-19 Research Consortium et al., 2020*; *Inciarte et al., 2020*; *Hadi et al., 2020*). Worse outcomes for PLWH tended to be in patients with low CD4 (*Hoffmann et al., 2021a*; *Dandachi et al., 2021*; *Braunstein et al., 2021*) and low absolute CD4 count was a risk factor for more severe disease (*Western Cape Department of Health in collaboration with the National Institute for Communicable Diseases, South Africa et al., 2021*).

HIV is known to interfere with protective vaccination against multiple pathogens (*Avelino-Silva et al., 2016*; *Carson et al., 1995*; *Cooper et al., 2011*; *Fuster et al., 2016*), typically as a consequence of sub-optimal antibody responses. In line with this, results from a South-African phase IIb trial of the Novavax NVX-CoV2373 vaccine, which uses a stabilised prefusion spike protein, showed 60% efficacy in HIV-uninfected individuals. However, overall efficacy dropped to 49% upon inclusion of PLWH (*Shinde et al., 2021*), although it is important to note that the numbers of PLWH in the study were very small. Nonetheless, there were more breakthrough cases in PLWH in the vaccine arm than the placebo arm.

An important consideration in infections in South Africa is the infecting variant, which in the second infection wave peaking January 2021 was predominantly the B.1.351 variant of concern (VOC) now designated as the Beta variant. In the current third infection wave it is predominantly the B.1.617.2 Delta variant. We and others have shown that the Beta variant has evolved the ability to escape neutralization by antibody responses elicited by earlier strains of SARS-CoV-2 or by vaccines based on those strains (*Cele et al., 2021*; *Wibmer et al., 2021*; *Garcia-Beltran et al., 2021*; *Hoffmann et al., 2021b*). Loss of vaccine efficacy of the AstraZeneca ChAdOx vaccine in South Africa was associated with this drop in neutralization capacity (*Madhi et al., 2021*). The second infection wave driven by Beta infections also showed increased mortality of hospitalized cases relative to the first infection wave (*Jassat et al., 2021b*).

What factors contributed to the evolution of the Beta variant in South Africa is yet unclear. One possibility is intra-host evolution in immunosuppressed PLWH with advanced HIV who are unable to

clear SARS-CoV-2 (*Karim et al., 2021*). There is also evidence that variants evolved other adaptations to the host in addition to those in the spike glycoprotein which lead to antibody escape and enhanced transmission. These include evolution of resistance to the host interferon response (*Guo et al., 2021*; *Thorne et al., 2021*), as well as enhanced cell-to-cell transmission (*Rajah et al., 2021*). Changes in the virus may make infection with some variants substantially different in disease course, transmission dynamics, and effect on PLWH relative to ancestral SARS-CoV-2 strains or other variants.

Here, we aimed to determine the effects of HIV on the immune response to SARS-CoV-2 infection in KwaZulu-Natal, South Africa. This is important because we need to better understand COVID-19 disease course and vaccine efficacy in this population, as well as the possible reasons for the emergence of variants which lead to immune escape from neutralizing antibodies. Our results indicate that infections in the Beta variant infection wave led to more severe disease in PLWH relative to HIV-negative participants. Higher severity was associated with a lower CD4 T cell count. Yet, the CD4 count recovered, indicating that these participants may not have had a low CD4 count when first exposed to SARS-CoV-2. In addition, there were changes in the response of immune cell subsets associated with SARS-CoV-2 infection in PLWH relative to HIV-negative participants in the first infection wave, even in the absence of a statistically significant increase in disease severity, indicating that HIV infection may modulate the immune response to SARS-CoV-2.

## Results

### HIV infection is associated with higher disease severity in the Beta variant infection wave

We initiated a longitudinal observational cohort study to enroll and track patients with a positive COVID-19 qPCR test presenting at three hospitals in Durban, South Africa. Patients presented due to either COVID-19 symptoms or because they were known contacts of a confirmed COVID-19 case.

All participants were initially admitted to a hospital facility, then discharged after varying periods and followed up as outpatients. Enrollment was between June 2020 and May 2021. Participants were followed up weekly for the first month post-enrollment, and at 3-month intervals thereafter. At each study visit, a blood sample and a combined nasopharyngeal and oropharyngeal swab was taken. The purpose of a combined swab was to maximize the detection probability by qPCR of SARS-CoV-2 in the upper respiratory tract. Blood was used to determine HIV status, HIV viral load, and cellular parameters such as the concentration of CD4 T cells and the NLR. We also tested the frequencies of more specific immune cell subsets by flow cytometry (only available for infection wave 1 samples).

Up to May 2021, 236 participants were enrolled in the study, for a total of 986 study visits (*Supplementary file 1*). All participants are assumed to be vaccinated with BCG in infancy in accordance with South African national guidelines. The majority of participants were female, possibly reflecting better linkage to care. Enrollment was a median 11 days post-symptom onset (*Supplementary file 2*). De-identified participant data used here are available as Source Data included in the supplementary materials.

Out of 236 study participants, 93 (39%) were PLWH (*Table 1*) and 89% of study participants were of African descent. PLWH were significantly younger than HIV uninfected participants. Hypertension, diabetes and obesity, known risk factors for more severe COVID-19 disease (*Zhou et al., 2020*; *the Northwell COVID-19 Research Consortium et al., 2020*), were common: Hypertension and obesity were present in 24%, and 42% of study participants respectively, a similar prevalence to that reported in the province of KwaZulu-Natal where this study was performed (*van Heerden et al., 2017*; *Malaza et al., 2012*). Diabetes prevalence in our study was 18%, compared to 13% reported for South Africa (*Federation, 2019*). Hypertension and diabetes were significantly lower in the PLWH group (*Table 1*). 28 or 30% of PLWH were HIV viremic at any point in the study. For individuals on ART, median ART duration was 9 years. ART regimen was determined by liquid chromatography with tandem mass spectrometry (LC-MS/MS) and was predominately efavirenz (EFV) based, with some participants transitioning to a dolutegravir (DTG) based regimen. In addition, there was a small subset of PLWH on a ritonavir boosted lopinavir (LPV/r) as well as other ART combinations. About 12% of PLWH had no detectable ART despite a clinical record of ART, or were ART naive

**Table 1.** Participant characteristics.

| | All (n=236) | HIV- (n = 143, 60.6%) | HIV+ (n=93, 39.4%) | Odds ratio (95% CI) | p-value |
|---|---|---|---|---|---|
| Demographics | | | | | |
| Age years, median (IQR) | 45 (35–57) | 49 (35–62) | 41 (35–50) | - | 0.003* |
| Male sex, n (%) | 82 (34.7) | 48 (33.6) | 34 (36.6) | 1.1 (0.7–2.0) | 0.68 |
| Current smoker, n (%) | 13 (5.5) | 4 (2.8) | 9 (9.7) | 3.7 (1.2 – $gt_{10}$) | 0.038 |
| Comorbidity, n (%) | | | | | |
| Hypertension#, n=235 | 57 (24.1) | 42 (29.4) | 15 (16.1) | 0.5 (0.2–0.9) | 0.023 |
| Diabetes | 42 (17.8) | 32 (22.4) | 10 (10.8) | 0.4 (0.2–0.9) | 0.024 |
| Obesity#, n=221 | 91 (42.3) | 64 (47.1) | 27 (29.0) | 0.6 (0.3–1.0) | 0.086 |
| Active TB | 10 (4.2) | 1 (0.7) | 9 (9.7) | >10 | 0.001 |
| History TB | 32 (13.6) | 3 (2.1) | 29 (31.2) | >10 | <0.0001 |
| HIV associated parameters | | | | | |
| HIV viremic, n (% of all HIV) | - | - | 28 (30.1) | - | - |
| Years ART, median (IQR) | - | - | 9.4 (3.9–13.2) | - | - |
| CD4 cells/µL median (IQR) n=221 | 633 (326–974) | 887 (534–1148) | 464 (200–702) | - | <0.0001* |
| CD4/CD8 | 1.2 (0.8–1.7) | 1.6 (1.2–2.1) | 0.8 (0.4–1.1) | - | <0.0001* |
| Disease severity, n (%) | | | | | |
| Asymptomatic | 33 (14.0) | 25 (17.5) | 8 (8.6) | 0.4 (0.2–1.0) | 0.058 |
| Ambulatory with symptoms | 128 (54.2) | 80 (55.9) | 48 (51.6) | 0.8 (0.5–1.4) | 0.59 |
| Supplemental oxygen | 62 (26.3) | 30 (21.0) | 32 (34.4) | 2.0 (1.1–3.5) | 0.024 |
| Death | 13 (5.5) | 8 (5.6) | 5 (5.4) | 1.0 (0.3–2.9) | >0.99 |
| COVID-19 treatment, n (%) | | | | | |
| Corticosteroids | 74 (31.2) | 47 (32.9) | 27 (29.0) | 0.8 (0.5–1.5) | 0.57 |
| Anticoagulants | 53 (22.5) | 35 (24.5) | 18 (19.4) | 0.7 (0.4–1.4) | 0.43 |
| Symptom, n (%) | | | | | |
| Sore throat | 88 (37.3) | 55 (38.5) | 33 (35.5) | 0.9 (0.5–1.5) | 0.68 |
| Runny nose | 53 (22.5) | 30 (21.0) | 23 (24.7) | 1.2 (0.7–2.3) | 0.53 |
| Cough | 153 (64.8) | 91 (63.6) | 62 (66.7) | 1.1 (0.7–2.0) | 0.68 |
| History of fever#, n=235 | 58 (24.7) | 29 (20.3) | 29 (31.2) | 1.8 (1.0–3.3) | 0.063 |
| Shortness of breath | 148 (62.7) | 87 (60.8) | 61 (65.6) | 1.2 (0.7–2.1) | 0.49 |

p-value calculated via 2-sided Fisher's Exact test, except for * which was calculated via Mann-Whitney U test. # Not including pregnancy or unable to be measured.

(*Supplementary file 3*). The absolute CD4 T cell count and the CD4 to CD8 T cell ratio was significantly lower in PLWH relative to HIV negative participants at enrollment. The incidence of active TB and the fraction of participants with a history of TB were much higher in the PLWH group (*Table 1*).

A minority of study participants (14%) were asymptomatic and presented at the hospital because of a close contact with a confirmed COVID-19 case. To include the asymptomatic participants in our analysis, we used time from diagnostic swab as our timescale, which was tightly distributed for symptomatic participants relative to symptom onset at a median of 3 to 4 days apart (*Supplementary file 2*).

The majority of participants in the study (54%) had symptoms but did not progress beyond mild disease, defined here as not requiring supplemental oxygen during the course of disease and convalescence. Twenty-six percent of participants required supplemental oxygen but did not die and 6% of participants died. Our cohort design did not specifically enroll critical SARS-CoV-2 cases. The requirement for supplemental oxygen, as opposed to death, was therefore our primary measure for disease severity.

There was a significant difference in the frequency of participants requiring supplemental oxygen (without subsequent death) between HIV-negative participants and PLWH (21% versus 34% respectively, odds ratio of 2.0 with 95% confidence intervals of 1.1–3.5, *Table 1*).

To determine if the fraction of participants requiring supplemental oxygen differed between the first infection wave and the Beta variant dominated second infection wave, we compared disease severity between the first infection wave (*Figure 1*, *Supplementary file 4*), and the second infection wave (*Figure 1*, *Supplementary file 5*). In the first infection wave, there was no significant difference in the fraction of participants requiring supplemental oxygen between HIV-negative and PLWH participants (*Supplementary file 4*, p=0.5). However, significantly more PLWH required supplemental oxygen in the second wave (*Supplementary file 5*, odds ratio of 4.0 with 95% CI of 1.6–10.4, p=0.005). Comparing within the HIV-negative and PLWH groups, there was only a moderate increase in the fraction of participants requiring supplemental oxygen between SARS-CoV-2 infection wave 1 and infection wave 2 in HIV-negative participants (19% to 25%) which was not significant (*Figure 1*). In contrast, the number of PLWH participants requiring supplemental oxygen more than doubled from 24% to 57% (p=0.0025, *Figure 1*).

To examine whether the differences in the requirement for supplemental oxygen in PLWH were because of differences in the level of HIV control between waves, we examined the fraction of timepoints where participants showed HIV viremia. We excluded low level viremia and set the threshold at VL > 200 HIV RNA copies/mL (*Ryscavage et al., 2014*). Furthermore, we determined whether ART was detectable in the blood by LC-MS/MS. Second wave participants had approximately twofold higher fraction of timepoints where HIV viremia was detected (*Figure 1—figure supplement 1A*). In agreement with this, the fraction of participants with no detectable ART in the blood was also about twofold higher (*Figure 1—figure supplement 1B*). These observations are consistent

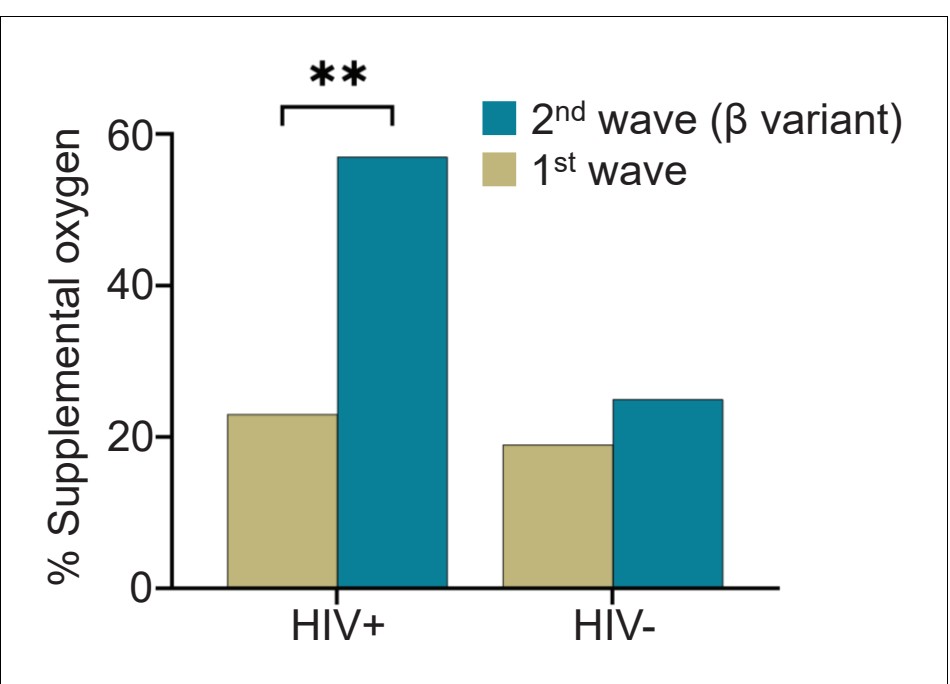

**Figure 1.** Fraction of PLWH and HIV-negative participants requiring supplemental oxygen during the first and the Beta variant dominated second infection waves. p=0.0025 by Fisher's Exact test.

The online version of this article includes the following figure supplement(s) for figure 1:

**Figure supplement 1.** Viremia and ART in PLWH in wave 1 versus wave 2.

**Figure supplement 2.** Effect of ART regimen on disease severity.

**Figure supplement 3.** Distribution of CD4 counts by HIV status.

**Figure supplement 4.** Viremia and ART in PLWH requiring versus not requiring supplemental oxygen.

**Figure supplement 5.** Dependence of time to SARS-CoV-2 clearance on CD4 count and HIV status.

with diminished suppression of HIV in second wave PLWH enrolled in this study. The specific HIV regimen had no discernible effect on disease severity (*Figure 1—figure supplement 2*).

We compared comorbidities and other characteristics between the PLWH and HIV negative participants on supplemental oxygen (*Table 2*). Strikingly, the median age of PLWH on supplemental oxygen was 21 years younger relative to HIV negative (41 versus 62, p=0.003). PLWH had significantly lower frequency of comorbidities which are usually associated with more severe COVID-19 disease: both hypertension (p=0.03) and diabetes (p=0.03) were lower. In contrast, the median CD4 T cell count across all study visits was lower in PLWH (277 versus 339), although this difference did not reach statistical significance (p=0.07). There was no significant difference in the fraction of participants treated with corticosteroids (p=0.2).

Interestingly, when comparing HIV-negative participants requiring supplemental oxygen to those not requiring supplemental oxygen (*Supplementary file 6*), those on supplemental oxygen were significantly older (62 versus 47 years, p=0.002), and had significantly higher frequency of hypertension (p=0.002) and diabetes (p=0.02). This differed from PLWH, where differences in age and comorbidities were not significant between PLWH requiring supplemental oxygen and those not (*Supplementary file 7*), although there was a trend to a higher frequency for hypertension (p=0.1).

HIV viremic participants showed lower CD4 counts relative to HIV suppressed or HIV negative participants (*Figure 1—figure supplement 3*). Surprisingly, there was no difference in either the fraction of HIV viremic timepoints or fraction of timepoints where ART was not detected in the blood between the group of PLWH requiring supplemental oxygen and the no supplemental oxygen group (*Figure 1—figure supplement 4*). We also analyzed the time of SARS-CoV-2 clearance as a function of CD4 count and HIV status and found that while participants with a low CD4 count (<200) showed a trend of longer time to SARS-CoV-2 clearance (p=0.11), HIV viremia had no effect (*Figure 1—figure supplement 5*). Hence, while the PLWH enrolled in the second wave had both worse control of HIV infection and had a higher fraction requiring supplemental oxygen, we did not observe that the PLWH requiring supplemental oxygen had a higher frequency of HIV viremia.

**Table 2.** Characteristics by HIV status of participants requiring supplemental oxygen.

| | All (n=68) | HIV- (n = 35, 51.5%) | HIV+ (n=33, 48.5%) | Odds ratio (95% CI) | p-value |
|---|---|---|---|---|---|
| Demographics | | | | | |
| Age years, median (IQR) | 51 (38–64) | 62 (47–66) | 41 (36–56) | - | 0.003* |
| Male sex, n (%) | 25 (36.8) | 12 (34.3) | 13 (39.4) | 1.2 (0.5–3.3) | 0.80 |
| Current smoker, n (%) | 2 (2.9) | 1 (2.9) | 1 (3.0) | 1.1 (<0.1 – >10) | $gt_{0.99}$ |
| Comorbidity, n (%) | | | | | |
| Hypertension | 26 (38.2) | 18 (51.4) | 8 (24.2) | 0.3 (0.1–0.8) | 0.026 |
| Diabetes | 17 (25.0) | 13 (37.1) | 4 (12.1) | 0.2 (0.1–0.8) | 0.025 |
| Obesity#, n=57 | 23 (40.4) | 11 (31.4) | 12 (36.4) | 1.8 (0.6–5.1) | 0.42 |
| Active TB | 6 (8.8) | 1 (2.9) | 5 (15.2) | 6.1 (0.9 – >10) | 0.10 |
| History TB | 16 (23.5) | 2 (5.7) | 14 (42.4) | 12.2 (2.7 – >10) | $lt_{0.001}$ |
| HIV associated parameters | | | | | |
| HIV viremic, n (% of all HIV) | - | - | 9 (27.3) | - | - |
| Years ART, median (IQR) | - | - | 11.6 (6.1–13.3) | - | - |
| CD4 cells/μL median (IQR) n=65 | 309 (170–545) | 339 (227–592) | 277 (134–461) | - | 0.072* |
| COVID-19 treatment, n (%) | | | | | |
| Corticosteroids | 43 (63.2) | 25 (71.4) | 18 (54.5) | 0.5 (0.2–1.3) | 0.21 |
| Anticoagulants | 31 (45.6) | 18 (51.4) | 13 (39.4) | 0.6 (0.2–1.6) | 0.34 |

p-value calculated via two-sided Fisher's Exact test, except for * which was calculated via Mann-Whitney U test. # Not including pregnancy or unable to be measured.

## SARS-CoV-2 has differential effects on CD4 count and the neutrophil to lymphocyte ratio between infection waves in PLWH

We next determined whether the increased disease severity in PLWH in infection wave two was reflected in the cellular immune response to SARS-CoV-2 infection. We therefore examined the CD4 count and NLR, both known to be strongly associated with disease severity. We used a three-point scale for disease severity, where 1: asymptomatic, 2: mild, and 3: supplemental oxygen (at any point in the study) or death. Death was merged with supplemental oxygen because of the small number of participants who died, and was not excluded in any of the subsequent analyses.

As expected, we observed a significant decrease in CD4 T cell count at the highest severity which included disease that required administration of supplemental oxygen and/or resulted in death (*Figure 2A*, see *Figure 2—figure supplement 1* for all data points and number of data points per graph).

We then asked whether PLWH in infection wave two showed different CD4 T cell responses to SARS-CoV-2. Since decreased CD4 count could be due to HIV infection alone, we separated the data into timepoints when SARS-CoV-2 was detectable by qPCR and after SARS-CoV-2 was cleared. Upon SARS-CoV-2 clearance, the immune response of convalescent participants should start the return to baseline, and differences due to SARS-CoV-2 should decrease and reflect HIV mediated effects only.

The CD4 counts in PLWH in infection wave 2 were lower during active SARS-CoV-2 infection relative to wave 1 (*Figure 2B*, median 172 versus 420 cells/μL, a decrease of 2.4-fold) and were below the 200 cells/μL clinically used threshold indicating a low CD4 count. However, CD4 counts for PLWH for both wave 2 and wave 1 recovered post-SARS-CoV-2 clearance (408 for wave 2 versus 584 cells/μL for wave 1), consistent with the low CD4 count in PLWH in wave 2 being SARS-CoV-2 induced. CD4 counts for both groups were substantially above the 200 cells/μL threshold after SARS-CoV-2 clearance. HIV-negative participants showed no or minor differences in CD4 counts between waves, although these minor differences showed significance due to the large number of participant timepoints for this group (*Figure 2C*).

The NLR had a remarkably similar pattern. An elevated NLR associated strongly with higher disease severity (*Figure 2D*). PLWH with active SARS-CoV-2 infection in wave 2 showed a twofold increase in the NLR relative to PLWH with active SARS-CoV-2 infection in wave 1 (*Figure 2E*). This difference declined to 1.2-fold once SARS-CoV-2 was cleared, consistent with differences in NLR being SARS-CoV-2 driven and not a result of other pathology in PLWH in wave 2. In contrast, the NLR was lower in HIV negative participants in wave 2 relative to wave 1 in the presence of SARS-CoV-2 (*Figure 2F*).

The observed recovery of the CD4 count may result from improved access to ART due to the hospital visit in wave 2. We therefore checked whether the fraction of HIV viremic participants decreased upon convalescence and whether there was an associated decrease in the number of PLWH with undetectable ART. We observed no significant differences in either viremia or fraction of PLWH with undetectable ART in either wave between timepoints which were SARS-CoV-2 positive and those that were negative (*Figure 2—figure supplement 1*). This indicates that the increase in the CD4 was not due to better linkage to care after the hospital visit but rather due to SARS-CoV-2 clearance.

## Differences in the frequencies and associations of immune cell subsets in PLWH and HIV-negative participants

To examine differences in immune cell subset associations between HIV-negative and PLWH participant groups, we conducted detailed phenotyping of immune cells using longitudinal fresh PBMC samples and correlated these to measured phenotypes and clinical parameters in both HIV-negative and PLWH groups (*Figure 3*; see *Figure 3—figure supplement 1* for gating strategies). We used established approaches for gating of cell subsets (*Sanz et al., 2019*; *Khodadadi et al., 2019*). This was only performed for the first wave participants, where cells were available for additional phenotyping by flow cytometry.

For HIV-negative participants, there were significant negative and positive correlations between CD4 T cell parameters, and between these and the CD8 T cell count and phenotypes (*Figure 3*, yellow box). There were negative correlations between CD4 and the CD8 CCR7+ T cell phenotype and

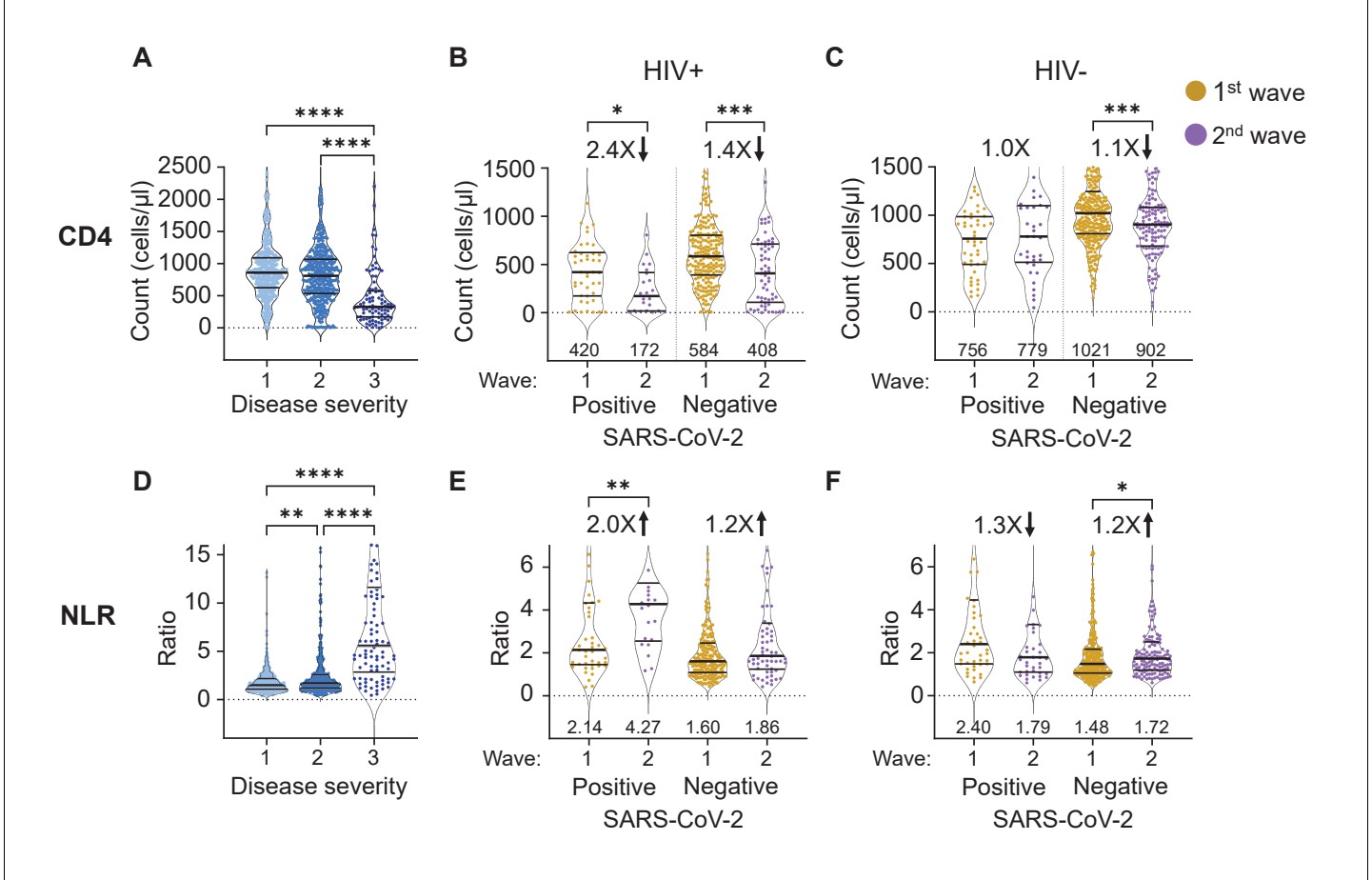

**Figure 2.** The differential effect of HIV on the CD4 count and neutrophil to lymphocyte ratio between waves. (A) The concentration of CD4 T cells in the blood in all participants in all infection waves and at all timepoints as a function of disease severity. Disease severity was scored as 1: asymptomatic, 2: mild, and 3: on supplemental oxygen or death. CD4 counts in PLWH (B) and HIV negative (C) participants in wave 1 versus wave 2 during active SARS-CoV-2 infection and after SARS-CoV-2 clearance. (D) Neutrophil to lymphocyte ratio (NLR) in the blood in all participants in all infection waves and at all timepoints as a function of disease severity. NLR in PLWH (E) and HIV negative (F) participants in wave 1 versus wave 2 during active SARS-CoV-2 infection and after SARS-CoV-2 clearance. SARS-CoV-2 positive indicates a timepoint where SARS-CoV-2 RNA was detected. Data shown as violin plots with median and IQR, with the median denoted below each plot. Fold-change in the second wave versus first wave is indicated, with arrow denoting direction of change. p-values are * <0.05; ** <0.01; *** < 0.001, **** < 0.0001 as determined by Kruskal-Wallis test with Dunn's multiple comparison correction or by Mann-Whitney U test. Plots scales were restricted to highlight changes close to the median.

The online version of this article includes the following figure supplement(s) for figure 2:

**Figure supplement 1.** The differential effect of HIV on the CD4 count and neutrophil to lymphocyte ratio between waves - full dataset and number of data points per plot.

**Figure supplement 2.** No significant increase in control of HIV infection at convalescence relative to active SARS-CoV-2 infection.

CD56+CD16+ NK cells (purple box). The fraction of NK cells positively correlated with the CXCR3 fraction of CD4 T cells, with HLA-DR on CD8 T cells, and with PD-1 on both cell types (purple box). In addition, there were correlations between CD8 T cell count and CD19 B cell parameters, such as fractions of naïve and memory B cells (red box). Interestingly, disease severity as well as the CD4/CD8 ratio showed correlations with B cell parameters, including the frequency of antibody secreting cells (ASC), which were lost in PLWH (orange box).

New correlations arose in PLWH, particularly involving CD8 T cells: CXCR3+ CD8 T cells were negatively correlated with disease severity but positively correlated with the CD4/CD8 ratio and the CD4 T cell count (*Figure 3*, black box). CD8 T cell activation (HLA-DR+) was correlated with several CD19+ B cell phenotypes (green box), and the plasma cell to plasmablast ratio, determined by CD138 expression, correlated with both CD4 and CD8 T cell phenotypes (blue box). In addition,

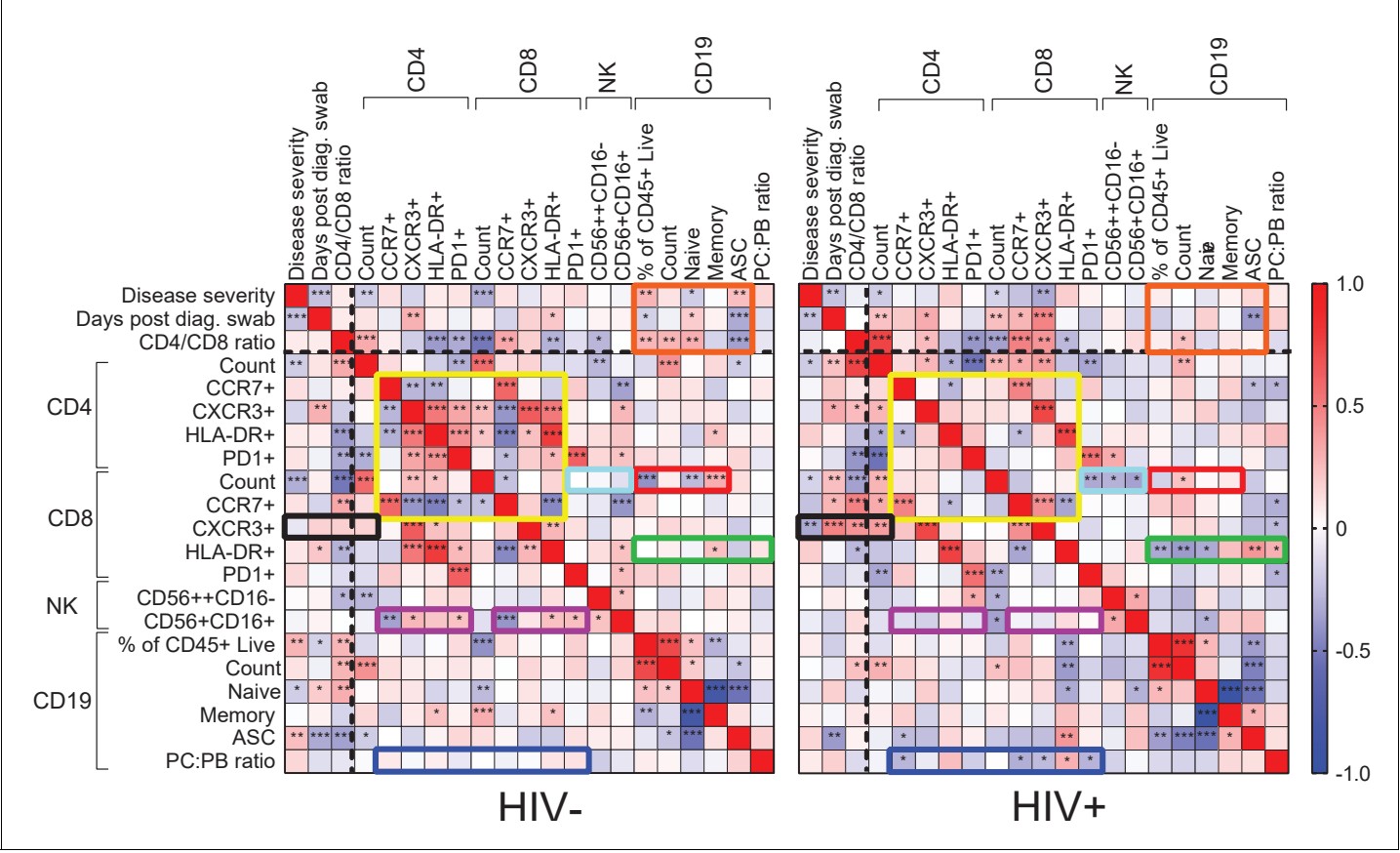

**Figure 3.** Immune cell and clinical correlates in HIV negative and PLWH groups. Spearman rank correlation values ($\rho$) are shown from red (1.0) to blue (−1.0). p-values per correlation are *< 0.5; **< 0.01; ***< 0.001. The number of matched pairs for HIV negative participants ranged from 77 to 229 and for PLWH from 48 to 164. Rectangles represent regions where a set of correlations is present in one group and absent in the other. Black dashed lines represent the divide between clinical and cellular parameters.

The online version of this article includes the following figure supplement(s) for figure 3:

**Figure supplement 1.** Gating strategy.

CD8 T cell count showed negative correlations with CD8 PD-1 and NK cell phenotypes only in PLWH (turquoise box).

Out of the set of markers examined, the combination of PD-1 and HLA-DR expression is linked to T cell activation (*Sauce et al., 2007*; *Vollbrecht et al., 2010*), while CXCR3 expression is essential to recruitment of T cells to tissues (*Groom and Luster, 2011*). We therefore asked whether these markers showed differences between HIV negative and PLWH in the first infection wave during the time participants were positive for SARS-CoV-2, despite there being no significant differences in disease severity in this wave. In CD8 T cells, we observed a significant decrease in the fraction of CXCR3 expressing cells in the blood compartment in PLWH relative to HIV-negative participants (*Figure 4A*). We also observed an increase in the fraction of PD-1+HLA-DR+ cells (*Figure 4B*). For CD4 cells, there was no significant decrease in the fraction of CXCR3+ cells although a decrease was apparent (*Figure 4C*). Similarly to CD8 T cells, there was an increase in PD-1+HLA-DR+ CD4 T cells in PLWH (*Figure 4D*). There was no difference between PLWH and HIV-negative participants in any cell/marker combination after SARS-CoV-2 clearance.

## Discussion

We observed that in our cohort, COVID-19 disease severity was higher in PLWH, consistent with some of the larger epidemiological studies (*Western Cape Department of Health in collaboration with the National Institute for Communicable Diseases, South Africa et al., 2021*; *Geretti et al.,*

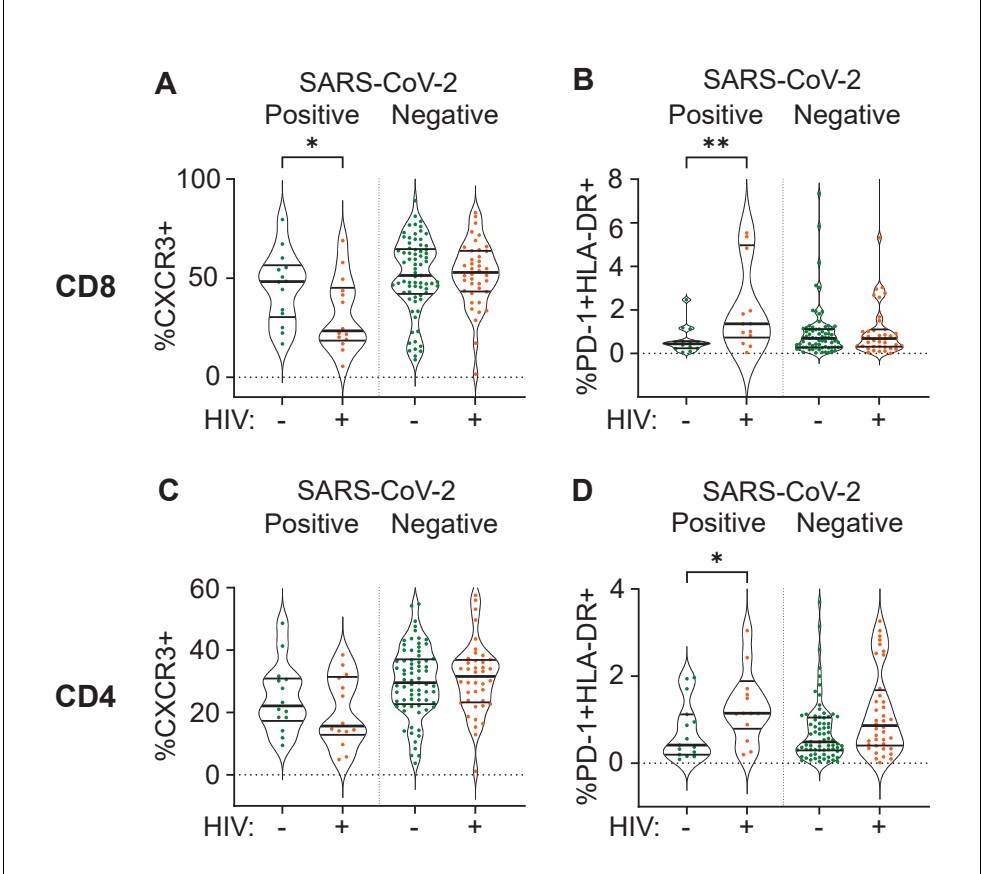

**Figure 4.** Differences between PLWH and HIV-negative participants in immune cell markers. Percent of CD8 T cells positive for CXCR3 (**A**) or double positive for HLA-DR and PD-1 (**B**). Percent of CD4 T cells positive for CXCR3 (**C**) or double positive for HLA-DR and PD-1 (**D**). Data is composed of 15 participant timepoints which were SARS-CoV-2+HIV-, 14 SARS-CoV-2+HIV+, 40 SARS-CoV-2-HIV+, and 74 SARS-CoV-2-HIV-, where SARS-CoV-2+ indicates SARS-CoV-2 RNA was detected in the upper respiratory tract. p-values for differences between PLWH and HIV-negative participants are * <0.05; ** <0.01; *** < 0.001, **** < 0.0001 as determined by the Mann-Whitney U test.

2021; *Bhaskaran et al., 2021*; *Tesoriero et al., 2021*; *Braunstein et al., 2021*; *Jassat et al., 2021a*), although in this study differences were detected in the frequency of participants requiring supplemental oxygen and not in mortality. Our cohort may not be a typical 'hospitalized cohort' as the majority of participants did not require supplemental oxygen. We therefore cannot discern effects of HIV on critical SARS-CoV-2 cases since these numbers are too small in the cohort. However, focusing on lower disease severity enabled us to capture a broader range of outcomes which predominantly ranged from asymptomatic to requiring supplemental oxygen. Understanding this part of the disease spectrum could be important since it may indicate underlying changes in the immune response which affect long-term quality of life and response to vaccines.

We observed a higher fraction of PLWH requiring supplemental oxygen relative to HIV negative participants in the second, Beta variant dominated SARS-CoV-2 infection wave in KwaZulu-Natal, South Africa. The odds ratio for requiring supplemental oxygen in the second wave for PLWH was 4.0 relative to HIV negative participants. The 95% confidence intervals were wide at 1.6–10.4, reflecting the relatively small number of participants. However, confidence intervals did not overlap one.

Consistent with HIV infection leading to more severe SARS-CoV-2 infection outcomes in our study is the much younger age of PLWH requiring supplemental oxygen relative to HIV negative participants (41 versus 63 years). PLWH on supplemental oxygen also had lower frequencies of hypertension and diabetes. Age, hypertension, and diabetes are risk factors for more severe COVID-19 disease (*Yang et al., 2020*; *Guan et al., 2020*; *Ambrosioni et al., 2021*; *Jassat et al., 2021a*), and

their absence may indicate that the more severe outcome is driven by another factor, with HIV infection being the simplest explanation.

The cause of the difference between waves in PLWH may be because PLWH enrolled in the second infection wave had worse suppression of HIV with ART: both the fraction of timepoints where viremia was detected and where ART was absent were about twofold higher and indeed were very high at about 40%. We therefore expected that this showed a direct link between HIV viremia and the requirement for supplemental oxygen during COVID-19 disease in PLWH. However, there was no difference in the frequency of viremia between those requiring supplemental oxygen and those not.

Furthermore, the substantial recovery of CD4 T cell counts in PLWH after SARS-CoV-2 clearance in wave 2 may be consistent with the Beta variant having more impact on the CD4 count relative to the ancestral SARS-CoV-2 strain infections in the first wave. A similar pattern was seen in the NLR, which was higher in wave two relative to wave 1 in PLWH with active SARS-CoV-2 infection, but then decreased to similar levels upon convalescence. The role of the Beta variant is supported by data showing extensive evolution, increasing the ability of Beta to escape the interferon response and result in more efficient viral cell-to-cell transmission (*Guo et al., 2021*; *Thorne et al., 2021*; *Rajah et al., 2021*). Beta variant hospitalizations also led to more deaths in South Africa (*Jassat et al., 2021b*). Therefore, the effect of the variant on PLWH in addition to HIV suppression status should be considered.

Our data detailing the SARS-CoV-2 response of more defined immune cell subsets in PLWH versus HIV negative participants is limited by the data only being available for the first infection wave. However, even in samples from that wave, there were multiple differences in correlations between cell subsets in PLWH relative to HIV negative participants, which may be another indication of differences in the immune response to SARS-CoV-2. We cannot deduce from these associations whether the differences could have an impact on disease severity. However, the fraction of CXCR3+ CD8 T cells decreased in the blood compartment and PD-1+HLA-DR+ CD8 and CD4 T cells increased. The increase in PD-1+HLA-DR+ T cells indicates T cell activation (*Sauce et al., 2007*; *Vollbrecht et al., 2010*) which associates with worse COVID-19 outcomes (*Chen et al., 2020b*). CXCR3 plays a key role in T cell homing to sites of inflammation and is activated by interferon-inducible ligands CXCL9, CXCL11, and CXCL10 (IP-10) (*Groom and Luster, 2011*; *Rodda et al., 2021*). A decrease in CXCR3 indicates either that T cells are less able to home to the site of infection, or that there is more inflammation in PLWH during SARS-CoV-2 infection and therefore more homing of the CXCR3+ CD8 T cells to tissues so that the fraction of CXCR3+ cells left in the blood decreases. Either way, the combination of these changes likely indicates either more pronounced SARS-CoV-2 infection or an impaired response in PLWH despite the similar infection outcomes in this wave.

In summary, PLWH showed increased disease severity mostly restricted to the second infection wave, where the Beta variant was dominant. Increased severity was associated with low CD4 T cell counts and high NLR which stabilized post-SARS-CoV-2 clearance in second wave infected PLWH to close to wave 1 PLWH values, arguing for a synergy between SARS-CoV-2 and HIV to decrease CD4 T cell numbers and increase the NLR rather than the status of HIV infection alone determining these parameters. More work is required to understand how these HIV related immune perturbations influence long-term immunity to SARS-CoV-2 infection and whether vaccine response will be affected.

## Materials and methods

### Ethical statement and study participants

The study protocol was approved by the University of KwaZulu-Natal Institutional Review Board (approval BREC/00001275/2020). Adult patients (>18 years old) presenting at King Edward VIII, Inkosi Albert Luthuli Central, or Clairwood Hospitals in Durban, South Africa, between June 2020 to May 2021, diagnosed to be SARS-CoV-2 positive as part of their clinical workup and able to provide informed consent were eligible for the study. Written informed consent was obtained for all enrolled participants.

## Clinical laboratory testing

An HIV rapid test and viral load quantification was performed from a 4 ml EDTA tube of blood at an accredited diagnostic laboratory (Molecular Diagnostic Services, Durban, South Africa) using the RealTime HIV negative1 viral load test on an Abbott machine. CD4 count, CD8 count, and a full blood count panel were performed by an accredited diagnostic laboratory (Ampath, Durban, South Africa). Depending on the volume of blood which was drawn, the CD8, CD4, and full blood count was not available for every participant, and numbers performed are detailed in the figure legends.

## qPCR detection of SARS-CoV-2

RNA was extracted from combined oropharyngeal and nasophryngeal swabs from 140 µl viral transport medium using the QIAamp Viral RNA Mini kit (cat. no. 52906, QIAGEN, Hilden, Germany) according to manufacturer's instructions, and eluted into 100 µl AVE buffer. To detect SARS-CoV-2 RNA, 5 µl RNA was added to the TaqPath 1-step RT-qPCR mastermix. 3 SARS-CoV-2 genes (ORF1ab, S and N) were amplified using the TaqPath COVID-19 Combo Kit and TaqPath COVID-19 CE-IVD RT-PCR Kit (ThermoFisher Scientific, Massachusetts, United States) in a QuantStudio 7 Flex Real-Time PCR system (ThermoFisher Scientific). Data was analyzed using the Design and Analysis software (ThermoFisher Scientific). For positive samples, Ct values are represented as the average of the Ct values of all three genes. A sample was scored positive where at least two out of the three genes were detected, and inconclusive if only one of the genes was detected.

## PBMC isolation and immune phenotyping by flow cytometry

PBMC were isolated by density gradient centrifugation using Histopaque 1077 (Sigma-Aldrich, St. Louis, Missouri, United States) and SepMate separation tubes (STEMCELL Technologies, Vancouver, Canada). For T cell and NK cell phenotyping, $10^6$ fresh PBMCs were surface stained in 50 microliter antibody mix with the following antibodies from BD Biosciences (Franklin Lakes, NJ, USA): anti-CD45 Hv500 (1:100 dilution, clone HI30, cat. 560777); anti-CD8 BV395 (1:50 dilution, clone RPA-T8, cat. 563795); anti-CD4 BV496 (1:25 dilution, clone SK3, cat. 564651); anti-PD1 BV421 (1:50 dilution, clone EH12.1, cat. 562516); anti-CXCR3 PE-CF594 (1:25 dilution, clone 1C6/CXCR3, cat. 562451). The following antibodies were from BioLegend (San Diego, CA, USA): anti-CD19 Bv605 (1:100 dilution, clone HIB19, cat. 302244); anti-CD16 Bv650 (1:50 dilution, clone 3G8, cat. 302042); anti-CD56 Bv711 (1:50 dilution, clone HCD56, cat. 318336); anti-CD3 Bv785 (1:25 dilution, clone OKT3, cat. 317330); anti-CXCR5 FITC (1:25 dilution, clone J252D4, cat. 356914); anti-HLA-DR PE (1:50 dilution, clone L243, cat. 307606); anti-CCR7 PerCP-Cy5.5 (1:25 dilution, clone G043H7, cat. 353220); anti-CD38 PE-Cy7 (1:25 dilution, clone HIT2, cat. 303516); anti-ICOS APC (1:25 dilution, clone C398.4A, cat. 313510) and anti-CD45RA AF700 (1:25 dilution, clone HI100, cat. 304120). PBMCs were incubated with antibodies for 20 min at room temperature. For B-cell phenotyping, the following antibodies were used: (all from BioLegend) anti-CD45 APC (1:25 dilution, clone HI30, cat. 304012); anti-CD3 Bv711 (1:50 dilution, clone OKT3, cat. 317328), anti-CD14 Bv711 (1:25 dilution, clone M5E2, cat. 301838); anti-CD19 Bv605 (1:50 dilution, clone HIB19, cat. 302244); anti-CD27 Hv500 (1:50 dilution, clone O323, cat. 302836); anti-CD38 PE-Cy7 (1:25 dilution, clone HIT2, cat. 303516) and anti-CD138 BV785 (1:25 dilution, clone MI15, cat. 356538). Cells were then washed twice in PBS and fixed in 2% paraformaldehyde and stored at 4˚C before acquisition on FACSAria Fusion III flow cytometer (BD) and analyzed with FlowJo software version 9.9.6 (Tree Star). Depending on the volume of blood which was drawn, full phenotyping was only available for participants where sufficient blood was available for the assay.

## Statistical analysis

Data is described with the non-parametric measures of median and interquartile range, and significance determined using the non-parametric Mann-Whitney U test for pairwise comparisons, Fisher Exact test for pairwise comparisons of frequencies, and the Kruskal-Wallis test with multiple comparison correction by the Dunn Method for comparisons involved more than two populations. All tests were performed using Graphpad Prism eight or Stata software.

## Acknowledgements

This work was supported by the Bill and Melinda Gates Investment INV-018944 to AS.

## Additional information

### Group author details

**COMMIT-KZN Team**

Moherndran Archary: Department of Paediatrics and Child Health, University of KwaZulu-Natal, Durban, South Africa; Kaylesh J Dullabh: Department of Cardiothoracic Surgery, University of KwaZulu-Natal, Durban, South Africa; Jennifer Giandhari: KwaZulu-Natal Research Innovation and Sequencing Platform, Durban, South Africa; Philip Goulder: Africa Health Research Institute and Department of Paediatrics, University of Oxford, Oxford, United Kingdom; Guy Harling: Africa Health Research Institute and the Institute for Global Health, University College London, London, United Kingdom; Rohen Harrichandparsad: Department of Neurosurgery, University of KwaZulu-Natal, Durban, South Africa; Kobus Herbst: Africa Health Research Institute and the South African Population Research Infrastructure Network, Durban, South Africa; Prakash Jeena: Department of Paediatrics and Child Health, University of KwaZulu-Natal, Durban, South Africa; Thandeka Khoza: Africa Health Research Institute, Durban, South Africa; Nigel Klein: Africa Health Research Institute and the Institute of Child Health, University College London, London, United Kingdom; Rajhmun Madansein: Department of Cardiothoracic Surgery, University of KwaZulu-Natal, Durban, South Africa; Mohlopheni Marakalala: Africa Health Research Institute and Division of Infection and Immunity, University College London, London, United Kingdom; Mosa Moshabela: College of Health Sciences, University of KwaZulu-Natal, Durban, South Africa; Kogie Naidoo: Centre for the AIDS Programme of Research in South Africa, Durban, South Africa; Zaza Ndhlovu: Africa Health Research Institute and the Ragon Institute of MGH, MIT and Harvard, Cambridge, United States; Kennedy Nyamande: Department of Pulmonology and Critical Care, University of KwaZulu-Natal, Durban, South Africa; Nesri Padayatchi: Centre for the AIDS Programme of Research in South Africa, Durban, South Africa; Vinod Patel: Department of Neurology, University of KwaZulu-Natal, Durban, South Africa; Theresa Smit: Africa Health Research Institute, Durban, South Africa; Adrie Steyn: Africa Health Research Institute and Division of Infectious Diseases, University of Alabama at Birmingham, Birmingham, United States

### Funding

| Funder | Grant reference number | Author |
| --- | --- | --- |
| Bill and Melinda Gates Foundation | INV-018944 | Alex Sigal |

The funders had no role in study design, data collection and interpretation, or the decision to submit the work for publication.

### Author contributions

Farina Karim, Conceptualization, Resources, Data curation, Formal analysis, Project administration; Inbal Gazy, Formal analysis, Methodology; Sandile Cele, Robert Krause, Mallory Bernstein, Data curation, Formal analysis; Yenzekile Zungu, Daniel Muema, Data curation; Khadija Khan, Yashica Ganga, Dirhona Ramjit, Project administration; Hylton Rodel, Ntombifuthi Mthabela, Investigation; Matilda Mazibuko, Willem Hanekom, Bernadett Gosnell, Tulio de Oliveira, Resources; Thumbi Ndung'u, Methodology; COMMIT-KZN Team, Richard J Lessells, Writing - review and editing; Emily B Wong, Conceptualization; Mahomed-Yunus S Moosa, Resources, Supervision; Gil Lustig, Formal analysis; Alasdair Leslie, Formal analysis, Supervision; Henrik Kløverpris, Formal analysis, Supervision, Investigation; Alex Sigal, Conceptualization, Funding acquisition, Writing - review and editing

## Author ORCIDs

Farina Karim (iD) https://orcid.org/0000-0001-9698-016X
Robert Krause (iD) https://orcid.org/0000-0003-1558-0397
Thumbi Ndung'u (iD) http://orcid.org/0000-0003-2962-3992
Richard J Lessells (iD) http://orcid.org/0000-0003-0926-710X
Alex Sigal (iD) https://orcid.org/0000-0001-8571-2004

## Ethics

Human subjects: The study protocol was approved by the University of KwaZulu-Natal Institutional Review Board (approval BREC/00001275/2020). Adult patients (>18 years old) presenting either at King Edward VIII, Inkosi Albert Luthuli Central or Clairwood Hospitals in Durban, South Africa, between June 2020 to May 2021, diagnosed to be SARS-CoV-2 positive as part of their clinical workup and able to provide informed consent were eligible for the study. Written informed consent was obtained for all enrolled participants.

## Decision letter and Author response

Decision letter https://doi.org/10.7554/eLife.67397.sa1
Author response https://doi.org/10.7554/eLife.67397.sa2

## Additional files

### Supplementary files

- Source data 1. Participant information.
- Supplementary file 1. Summary of case visits.
- Supplementary file 2. Timing of enrollment in PLWH and HIV negative participants.
- Supplementary file 3. ART regimen in PLWH as determined by LC-MS/MS.
- Supplementary file 4. Infection wave 1 COVID-19 disease severity by HIV status.
- Supplementary file 5. Infection wave 2 COVID-19 disease severity by HIV status.
- Supplementary file 6. Comparison between HIV negative participants requiring and not requiring supplemental oxygen.
- Supplementary file 7. Comparison between PLWH requiring and not requiring supplemental oxygen.
- Transparent reporting form

## Data availability

All data generated or analysed during this study are included in the manuscript and supporting files.

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
