## [Decision Letter]

**Acceptance summary:**

In this work the authors perform extensive immune profiling of HIV infected and uninfected persons through two waves of SARS-CoV-2 infection with different dominant variants overtime regionally in South Africa. A limitation is the minimal characterization of viral specific T cells. Overall this work will be relevant to our understanding the effect of COVID-19 in people with HIV, long term outcomes and also improve our understanding of vaccine designs.

**Decision letter after peer review:**

Thank you for submitting your article "HIV status alters immune cell dynamics in response to SARS-CoV-2 infection" for consideration by *eLife*. Your article has been reviewed by 3 peer reviewers, one of whom is a member of our Board of Reviewing Editors, and the evaluation has been overseen by Miles Davenport as the Senior Editor. The reviewers have opted to remain anonymous.

*Reviewer #1 (Recommendations for the authors):*

The Authors present results of a longitudinal observational cohort aimed at documenting the immunological changes across 124 participants during the first wave of the COVID-19 epidemic in KwaZulu Natal in South Africa in 2020. The goal of the study is centered on the ineffective vaccine responses seen in HIV infected persons and by assessment of the immune responses in HIV+/COVID-19+ co-infected HIV-/COVID-19+ patients they propose to ultimately inform mechanisms of vaccine efficacy likely against COVID-19 in this population. Several outcomes of interest restricted to effector immune populations are presented from COVID-19 participants from 2 hospital sites. The authors describe few clinical outcome differences between the groups that is in line with previous studies. However they do uncover and document a pronounced CD8 T cell expansion, evidence of delayed tissue homing of CXCR3 T cells and lower expansion of antibody secreting B cells in COVID-19 and HIV coinfected group compared to the HIV uninfected group.

The manuscript is clear and well written and the study is moderately novel in that immunological studies of HIV and COVID-19 co-infection have been reported though not in the sub-region studied. The methods appear sound. The introduction of vaccines for COVID-19 and the emergence of variants in South Africa and how they may impact PLWH is well discussed making the findings presented a good reference backdrop for future assessment. Good literature review is also presented. Specific suggestions for improving the manuscript are as follows:

Commendable that SARSCoV-2 viral load measure were assessed however the relationship with immune parameters among the viremic populations would be informative.

Main concern about the study is the omission of data on severe cases of COVID-19 limit the full interpretation of the data. Any deaths recorded in both groups is unclear. An explanation was not adequately provided on these omissions. Alternatively, the objective should explicitly state this study was limited to non-critical cases.

Figure 1 Disease severity should be included a table in the actual figure.

There are several missing references on past studies on HIV and SARCoV2 infection.

The authors should consider some description of the ART regimen should be included or mentioned as % of class type in Table 1 given interest of some regimens may have some anti-COVID utility.

It is unclear if some patients were hospitalized or non-hospitalized.

*Reviewer #2 (Recommendations for the authors):*

The manuscript can improve by addressing some of these observations.

A. Text: the manuscript is well written but please move all the references, including link and hyperlinks, in the proper section.

B. Clarity in study design: adding a figure with the study design which includes number of participants and timeline would be helpful for the readers and reduce confusion. For example, Figure S1C states 5 follow up but the data have been grouped in the other results in 4 time points.

C. Figures and reference to figures please references in the manuscript specific panels (avoid referencing to S1 when the text refers to S1A). Please name all graphs: in the manuscript is difficult to follow when a specific panel (for example Figure 1B) contains 4 graphs. Please review Y axes titles and add information when needed. Figure S2 and S7 are not referenced enough in the manuscript.

D. Virologic data: please explain the choice to combine oral and nasal swabs and analyze them together. Did all participants receive both nasal and oral swabs? If yes, please state it.

E. Flow cytometry data: please reference to previous studies that guided the definition of ASCs using CD138. Why has CD21 not been used to define naive b cells? In Figure S3 there is an error in the gating of CCR7/CXCR3: the figure seems to refer to the same specimen and it seems that the two populations are complimentary but the sum of it is 104%.

*Reviewer #3 (Recommendations for the authors):*

My concerns lie with the immunological assays and interpretation of those results. Specifically, the choice of markers is rather limited, and thus the identities assigned to the subsets may not be quite accurate. For example, PD-1 alone is not a marker of exhausted cells, since activated cells also express PD-1, and exhausted cells are typically characterized by a combination of surface markers, of which PD-1 is one. Likewise HLA-DR alone would not be typically used to assign a cell as "activated." You may wish to engage a reviewer with further expertise in cellular immunology to comment on this, but in my assessment, the flow markers used do not justify the labels given to cell subsets.

I am also not entirely clear on the meaning of the ASC data. How would this relate to actual antibody (IgG) levels, and why not just (or in addition) measure these in patient serum?

Are the CD8 cells anti-viral? Could you do an interferon γ release assay using stored PBMC to answer this question?

In figure 1, I find panels C and D do not add value to what is relayed by panel B.

I question the validity of using the metric "% above normal," since "normal" % and absolute numbers of CD4 or CD8 in PLWH are inherently deranged from what would be expected in uninfected (on whom "normal" values are based).

For figure 4, do you account for multiple comparisons when calculating these p-values?

---

## [Author Response]

Reviewer #1 (Recommendations for the authors):The Authors present results of a longitudinal observational cohort aimed at documenting the immunological changes across 124 participants during the first wave of the COVID-19 epidemic in KwaZulu Natal in South Africa in 2020. The goal of the study is centered on the ineffective vaccine responses seen in HIV infected persons and by assessment of the immune responses in HIV+/COVID-19+ co-infected HIV-/COVID-19+ patients they propose to ultimately inform mechanisms of vaccine efficacy likely against COVID-19 in this population. Several outcomes of interest restricted to effector immune populations are presented from COVID-19 participants from 2 hospital sites. The authors describe few clinical outcome differences between the groups that is in line with previous studies. However they do uncover and document a pronounced CD8 T cell expansion, evidence of delayed tissue homing of CXCR3 T cells and lower expansion of antibody secreting B cells in COVID-19 and HIV coinfected group compared to the HIV uninfected group.

In the course expanding the study to the β infection wave and doing the revision it emerged that there was a difference between PLWH and HIV negative participants in terms of Covid-19 outcomes in the second wave. Over both infection waves, this is a moderate effect (odds ratio 2) but 95% CI do not overlap 1. In wave 2, the effect is strong (odds ratio 4). The expanded cohort is now presented in a modified Table 1.

Wave 2 specific data is presented in Supplementary file 5.

The striking difference between waves is shown in Figure 1.

The characteristics of participants requiring supplemental oxygen are different between HIV negative and PLWH. HIV negative participants have what are well known risk factors: older age and co-morbidities. This is not the case in PLWH, which seems to indicate that the driver for more severe outcome is different. The information is summarized in a new Table 2.

Therefore, we think that the clinical outcome analysis has been strengthened and is now important.

The manuscript is clear and well written and the study is moderately novel in that immunological studies of HIV and COVID-19 co-infection have been reported though not in the sub-region studied.

As discussed above, we think the novelty of the study has increased. In the previous iteration we did not have a clear result to investigate.

The methods appear sound. The introduction of vaccines for COVID-19 and the emergence of variants in South Africa and how they may impact PLWH is well discussed making the findings presented a good reference backdrop for future assessment. Good literature review is also presented. Specific suggestions for improving the manuscript are as follows:Specific Concerns:Commendable that SARSCoV-2 viral load measure were assessed however the relationship with immune parameters among the viremic populations would be informative.

We now use SARS-CoV-2 detection as our main measure to separate participants during the active stage of disease (SARS-CoV-2+) and those who have cleared the virus (SARS-CoV-2-, convalescence). We do this for the CD4 count and NLR (new Figure 2) to show that, as the reviewer suggests, immune cell frequencies in PLWH in wave 2 are more strongly affected, and move in directions associated with higher disease severity, relative to PLWH in wave 1. Upon SARS-CoV-2 clearance, differences between waves are greatly reduced.

Moreover, we perform a similar analysis for more detailed phenotypic markers (CXCR3 and HLA-DR/PD-1, analysis only available for wave 1 samples) and observe again that during active infection these markers are indicating more pronounced SARS-CoV-2 infection in PLWH. This is shown in a new Figure 4.

Main concern about the study is the omission of data on severe cases of COVID-19 limit the full interpretation of the data. Any deaths recorded in both groups is unclear. An explanation was not adequately provided on these omissions. Alternatively the objective should explicitly state this study was limited to non-critical cases.

Death has now been added to Table 1 under the “Disease severity” subheading. The number of participants who have died, at 13, is relatively small. We did not limit the study to non-critical cases. Our main measure of severity is supplemental oxygen. This is stated in the Results, line 110-112:

“Our cohort design did not specifically enroll critical SARS-CoV-2 cases. The requirement for supplemental oxygen, as opposed to death, was therefore our primary measure for disease severity.”

This is justified in the Discussion, lines 236-242:

“Our cohort may not be a typical 'hospitalized cohort' as the majority of participants did not require supplemental oxygen. We therefore cannot discern effects of HIV on critical SARS-CoV-2 cases since these numbers are too small in the cohort. However, focusing on lower disease severity enabled us to capture a broader range of outcomes which predominantly ranged from asymptomatic to supplemental oxygen, the latter being our main measure of more severe disease. Understanding this part of the disease spectrum is likely important, since it may indicate underlying changes in the immune response which could potentially affect long-term quality of life and response to vaccines.”

Figure 1 Disease severity should be included a table in the actually figure

We present a figure of our main measure of disease severity, which is the requirement for supplemental oxygen, as new Figure 1.

There are several missing references on past studies on HIV and SARCoV2 infection.

We now include 18 references on Covid-19/HIV (refs 33-50).

The authors should consider some description of the ART regimen should be included or mentioned as % of class type in Table 1 given interest of some regimens may have some anti-COVID utility.

ART regimen is now listed in Supplementary File 3: ART regimen in PLWH as determined by LC-MS/MS. We include a new Figure 1—figure supplement2 showing that regimen type has little bearing on Covid-19 disease severity.

It is unclear if some patients were hospitalized or non-hospitalized.

All participants were initially admitted to a hospital facility, then discharged after varying periods and followed up as outpatients. This is now stated in the Results, lines 74-75.

Reviewer #2 (Recommendations for the authors):The manuscript can improve by addressing some of these observations.A. Text: the manuscript is well written but please move all the references, including link and hyperlinks, in the proper section.

The hyperlinks are now references.

B. Clarity in study design: adding a figure with the study design which includes number of participants and timeline would be helpful for the readers and reduce confusion. For example, Figure S1C states 5 follow up but the data have been grouped in the other results in 4 time points.

On lines 75-76 of the Results we now state:

“Participants were followed up weekly for the first month post-enrollment, and at 3 month intervals thereafter. Up to May 2021, 236 participants were enrolled in the study, for a total of 986 study visits (Supplementary File1).”

“Supplementary File 1: Summary of case visits” lists how many participants were captured per timepoint. Given the changing pandemic, the study is ongoing and does not have a defined endpoint.

C. Figures and reference to figures please references in the manuscript specific panels (avoid referencing to S1 when the text refers to S1A). Please name all graphs: in the manuscript is difficult to follow when a specific panel (for example Figure 1B) contains 4 graphs. Please review Y axes titles and add information when needed. Figure S2 and S7 are not referenced enough in the manuscript.

This has been done.

D. Virologic data: please explain the choice to combine oral and nasal swabs and analyze them together. Did all participants receive both nasal and oral swabs? If yes, please state it.

We used a combined nasal and oral swab to maximize SARS-CoV-2 yield. This is now stated in the Results section, lines 77-79:

“The purpose of a combined swab was to maximize the detection probability by qPCR of SARS-CoV-2 in the upper respiratory tract.”

E. Flow cytometry data: please reference to previous studies that guided the definition of ASCs using CD138.

We include a review for this gating strategy as reference 72: Khodadadi, L., Cheng, Q., Radbruch, A. and Hiepe, F. The maintenance of memory plasma cells. Frontiers in immunology 10,721 (2019).

Why has CD21 not been used to define naive b cells?

To identify the canonical parent B cell populations CD27 and CD38 are usually sufficient, or alternatively CD27 and IgD. We add a review as reference 71: Sanz, I., Wei, C., Jenks, S. A., Cashman, K. S., Tipton, C., Woodruff, M. C., Hom, J. and Lee, F. Challenges and opportunities for consistent classification of human B cell and plasma cell populations. Frontiers in immunology10,2458 (2019).

CD21 is a critical marker not necessarily for the classification/definition of the parent populations but rather to assess the activation of these parent populations and would therefore be used in sub-gating.

In Figure S3 there is an error in the gating of CCR7/CXCR3: the figure seems to refer to the same specimen and it seems that the two populations are complimentary but the sum of it is 104%.

This has been corrected.

Reviewer #3 (Recommendations for the authors):My concerns lie with the immunological assays and interpretation of those results. Specifically, the choice of markers is rather limited, and thus the identities assigned to the subsets may not be quite accurate. For example, PD-1 alone is not a marker of exhausted cells, since activated cells also express PD-1, and exhausted cells are typically characterized by a combination of surface markers, of which PD-1 is one. Likewise HLA-DR alone would not be typically used to assign a cell as "activated." You may wish to engage a reviewer with further expertise in cellular immunology to comment on this, but in my assessment, the flow markers used do not justify the labels given to cell subsets.

We have taken the reviewer’s suggestion and combined PD-1 and HLA-DR and added references showing this combination is used to detect activated T cells. We also include CXCR3 and add references that this is a marker for T cell homing. The result is a new Figure 4, which shows that in PLWH there is increased activation (PD-1+ and HLA-DR+) and decreased fraction of CXCR3 CD8 T cells, indicating either that T cells are less able to home to the site of infection, or that there is more inflammation in PLWH during Covid-19 and therefore more homing of the CXCR3+ CD8 T cells to tissues so that the fraction of CXCR3+ cells left in the blood decreases.

I am also not entirely clear on the meaning of the ASC data. How would this relate to actual antibody (IgG) levels, and why not just (or in addition) measure these in patient serum?

We have removed this data.

Are the CD8 cells anti-viral? Could you do an interferon γ release assay using stored PBMC to answer this question?

We do not have an IGRA assay set up. It is an important follow-up question but we cannot address it in this work.

In figure 1, I find panels C and D do not add value to what is relayed by panel B.

We have removed this figure.

I question the validity of using the metric "% above normal," since "normal" % and absolute numbers of CD4 or CD8 in PLWH are inherently deranged from what would be expected in uninfected (on whom "normal" values are based).

We have removed this figure.

For figure 4, do you account for multiple comparisons when calculating these p-values?

A two-tailed nonparametric Spearman correlation with 95% confidence interval was used, with r calculated for every data set. These correlation matrices are not usually corrected for multiple hypothesis and are used to see trends across multiple parameters.